# Impact of Up-Scheduling Medicines on Pharmacy Personnel, Using Codeine as an Example, with Possible Adaption to Complementary Medicines: A Scoping Review

**DOI:** 10.3390/pharmacy8020065

**Published:** 2020-04-15

**Authors:** Kristenbella AYR Lee, Joanna E. Harnett, Carolina Oi Lam Ung, Betty Chaar

**Affiliations:** 1School of Pharmacy, The University of Sydney School of Pharmacy, Sydney 2006, Australia; klee7377@uni.sydney.edu.au (K.A.L.); joanna.harnett@sydney.edu.au (J.E.H.); 2State Key Laboratory of Quality Research in Chinese Medicine, Institute of Chinese Medical Sciences, University of Macau, Macau SAR, Macau 999078, China; carolinaung@um.edu.mo

**Keywords:** scheduling, complementary medicines, dietary supplements, regulation

## Abstract

Within Australia, vitamins, minerals, nutritional supplements, essential oils, and homoeopathic and herbal preparations are collectively termed and regulated as Complementary Medicines (CMs) by the Australian Therapeutic Goods Administration (TGA). CMs are predominantly self-selected through a pharmacy, providing pharmacy personnel an opportunity to engage with the public about their CM use. CMs are currently non-scheduled products in Australia. This review aimed to summarize the literature reporting the potential effect on pharmacies if scheduling of CMs was adopted, using codeine as an example. A scoping review methodology was employed. Seven databases were searched to identify four key concepts, including: CMs, scheduling and rescheduling, codeine, and pharmacists. Seven studies were included for analysis. The majority of the literature has explored qualitative studies on the perception and opinion of pharmacists in relation to the up-scheduling of codeine. The case of codeine illustrates the possible impact of up-scheduling. If CMs were to be up-scheduled, the accessibility of CMs would be limited to the pharmacy providing a role for pharmacy personnel, including both pharmacists and pharmacy technicians, to counsel on CM use. However, careful collaboration and consideration on how such a regulatory change would impact other key-stakeholders, including CM practitioners, requires both a strategic and collaborative approach.

## 1. Introduction

Within Australia, vitamins, minerals, nutritional supplements, essential oils, and homoeopathic and herbal preparations are collectively termed and regulated as Complementary Medicines (CMs) by the Australian Therapeutic Goods Administration (TGA) [1]. The prevalent use of CMs has steadily increased over the last two decades [2,3]. An estimated 53% of CMs are self-selected and mainly accessed through pharmacy outlets, providing pharmacists, technicians, and sales assistants an opportunity to engage with the public about their CM use [4,5]. CMs are currently non-scheduled products in Australia. However, as they are provided from within pharmacy premises. According to the principles of professional ethics clearly articulated in the Code of Ethics for Pharmacists [6], pharmacists and pharmacy personnel are expected to counsel patients and provide sufficient information to ensure safe and appropriate use, and pharmacists are expected to adopt an engaged professional role in relation to CMs provision [4,5]. This is particularly important in light of the growing body of evidence reporting potential side effects and drug-CM interactions that can impact patient safety and clinical outcomes of therapy. A well-established drug-herb interaction is *Hypericum perforatum* (St John’s Wort, or SJW) [7], which is used in the management of mild to moderate depression [7]. In Australia, St John’s Wort is easily accessed over-the-counter (OTC). Hyperforin, an active constituent of John’s Wort, is known to induce the cytochrome P-3A4 (CYP3A4) enzyme and the drug transporter p-glycoprotein, which are involved in the metabolism of many medications [8,9]. Concurrent use of SJW with these CYP3A4-metabolized medicines is well documented as resulting in significant interactions associated with patient harm [10].

Despite this, there are reports that barriers exist in the pharmacy environment which prevent pharmacists and pharmacy personnel from adopting professional standards regarding CMs. These barriers appear to be: time constraints, limited resources within the pharmacy, and a lack of knowledge about CMs [11]. Some pharmacists report that CMs were secondary to their primary concerns in patient care. They considered these products as being mostly “retail” products, available through a range of retail outlets in addition to pharmacies, indicating the uncertainty about whether they should assume professional responsibility for ensuring the appropriate and safe use of widely available CMs [11]. Consumers’ lack of respect towards the potential safety issues with the use of CMs was suggested to be yet another challenge experienced by pharmacists when trying to provide professional care related to CMs. To some consumers, as reflected by some pharmacists, having the CMs available at different retail outlets might indicate an “assurance of safe”; thus, pharmacists’ advice or intervention was not needed [11]. Whilst there appears to be consensus among pharmacists and associated stakeholders about the responsibility of the pharmacist to actively supervise provision of CMs, as is legally required in the case of ‘Pharmacist-Only’ (Schedule 3 in Australia) and ‘Prescription-Only’ (Schedule 4 in Australia) medicines [5], as well as clearly mentioned in the Code of Ethics for Pharmacists, it is unclear whether such professional responsibility could be extended to CMs in their current status; or should up-scheduling be considered?

Scheduling of medicines is one of the global key pillars of the pharmacy practice business model, with decisions to reschedule or up-schedule medicines made in an attempt to improve medicine use at a population level [12]. In Australia, the scheduling of medications is assigned according to the appropriate level of safety and control over the accessibility and availability of medicines. The aim of rescheduling is to safeguard the health and safety of the public. There are several schedules within the Australian classification system. Schedule 2 medications, also known as “Pharmacy Medicines”, are restricted to pharmacies and can be managed by non-pharmacist staff within the pharmacy. Provision of Schedule 3 medications (“Pharmacist-only Medicines”) require a pharmacist to engage in a consultation session with the patient to ascertain if there is a therapeutic need for the medication and to advise on its use [12]. Provision of “Prescription-only” medicines are classified under Schedule 4 and require a prescriber to fill a prescription for it to be dispensed by the pharmacist.

One example of the complexities and impact of rescheduling was the rescheduling in Australia of codeine-containing products (CCPs) in 2010, which was a very controversial topic at the time. Some low dose CCPs were previously scheduled at Schedule 2 or over-the-counter (OTC). However, with intense scrutiny of the misuse of opioid-containing medicines, and the rising incidence of addiction on these medicines in Australia, on the 1 May 2010, regulatory changes were made to minimize access and misuse, and all low dose CCPs were up-scheduled from Schedule 2 to Schedule 3 [13]. In Australia, patients could therefore no longer self-select CCPs, and the pharmacist was required to have some level of consultation with the patient before supply of the product. Being a Pharmacist-Only Medicine implied that the pharmacist had all professional and legal responsibilities for the supply of the Schedule 3 medicine, and as such, up-scheduling impacted on the practice of community pharmacists all around Australia [13]. Inadvertently, however, pharmacists became involved in the supply and identification of misuse of CCPs. Thus, despite the rescheduling in 2010, the rate of misuse did not decline with heightened pharmacists’ involvement. According to Cairns et al., even after the change in drug scheduling, phone calls made to New South Wales Poisons Information Centre (NSWPIC), coded under codeine-misuse, continued to rise [14]. Therefore, in December 2016, the TGA announced that CCPs were to undergo another round of up-scheduling to make them unavailable OTC or pharmacist-only, due to the safety issues of codeine. Accordingly, codeine was up-scheduled to Schedule 4 (prescription only) by the TGA on 1 February 2018, in an effort to address the increasing use and/or misuse of pharmaceutical opioids, particularly in relation to codeine abuse. The action of regulatory up-scheduling was therefore undertaken in the interest of patient safety and to minimize the risks of drug dependence and toxicity [14].

While the safety and implications of most CMs is incomparable to CCPs, the role of scheduling in relation to the accessibility of products and professionalism of pharmacists is worthy of discussion. To date, the scheduling of CMs has not even been considered or explored. Despite rising consumption and current safety concerns around some CMs, CMs remain unscheduled. And while CMs are clearly not drugs of abuse, there is an increasing demand from professional organizations, and within the literature, advocating for pharmacists and pharmacy personnel to adopt heightened professional duties as related to the supply of CMs [15]. According to the position statement issued by the Pharmaceutical Society of Australia, “*pharmacists are recommended to assume the responsibility of providing sound evidence-based advice to assist consumers in making informed decisions regarding CMs*” [16]. In the United States of America, where CMs are also over the counter, the position statement published by the American Society of Health System Pharmacists also urged pharmacists to integrate awareness of patients’ use of dietary supplements into everyday practice and to increase efforts to prevention of interactions between these products and prescriptions medicines [17]. This responsibility may be aided by community workforce personnel, such as pharmacy assistants, who, if trained appropriately, may direct requests for CMs to the pharmacist in charge, where necessary.

The Australian Industry and Skills Committee published a report on community pharmacy in April 2020 [18], stating the following: “*The Community Pharmacy sector plays an important role in the Health Care sector through the supply to the general public of prescription-based medicine, non-prescription-based medicine when permitted, and a range of information and health care services … As a result, the community pharmacy sector is pivotal in reducing the demand and burden on primary health care facilities … The Community Pharmacy sector includes a workforce of 41,400 pharmacy sales assistants and generated $18.4 billion in revenue in the 2018-19 period, up from $16.3 billion in 2016-17.*” And according to Australian government statistics, approximately 59% of the community pharmacy workforce is comprised of pharmacy assistants, many of whom undergo specific training to enable them to undertake tasks allocated to them in pharmacy [19]. As most requests for CMs come from the OTC areas (or shelves) within a typical community pharmacy, usually allocated to oversight of pharmacy assistants, there may be an important role for pharmacy personnel, such as pharmacy assistants, to support the pharmacist with streamlining of requests for CMs in a community pharmacy. This role would be particularly effective if CMs were to be scheduled in Australia.

We hypothesize that the scheduling of CMs to enable holding pharmacists and pharmacy personnel responsible for oversight of the supply of CMs could, theoretically at least, be a reasonable approach to enhance safe and appropriate use of self-selected CMs [11]. It would therefore follow that we seek to explore the literature for evidence of the impact of re-scheduling of other medicines in Australia. Hence, this is an exploratory study aimed to investigate the potential effect on pharmacy practice if scheduling of CMs was adopted, using the case of up-scheduling of codeine in Australia as an example.

## 2. Methodology

Arksey and O’Malley’s scoping review framework was adopted in our literature review [20]. The scoping review methodology involves five different stages: (i) identifying a research question, (ii) identifying relevant studies, (iii) the selection of the studies, (iv) charting of data, and (v) summarizing and reporting of the results.

This scoping review was conducted in accordance to the Preferred Reporting Items for Systematic Reviews and Meta-Analyses (PRISMA) statement [21]. A PRISMA chart was constructed (Figure A1). The databases used in the literature search were: Cumulative Index to Nursing and Allied Health Literature (CINAHL), Embase, Medline, PubMed, Scopus, Google Scholar and Web of Science. Our search strategy can also be found in Figure A2.

Studies that revolved around four concepts: (i) CMs, (ii) Scheduling and rescheduling, (iii) Codeine and (iv) Pharmacist, which were published in the English language between 10 March 2014 and 10 March 2019, were included. A timeframe of five years was selected due to the increasing trend of the use of CMs by consumers over this period [22]. Terminologies used in the search strategy for the four respective concepts are shown below:
CMs: Complementary medicines, natural medicines, dietary supplements, vitamins, minerals, herbal supplements, homeopathic medicines, complementary therapy, CMs, and aromatherapy oils.Scheduling and rescheduling: regulations, up-schedule.Codeine.Pharmacist: Pharmacist, retail pharmacy, community pharmacy, pharmacy management and pharmacist autonomy.


Research articles that did not follow IMRAD (Introduction, Methods, Results, and Discussion) format were excluded from the study. Letters, editorials, and commentaries were also excluded. Search term criteria and terminologies were agreed upon among the four authors, and one author performed the literature search. The same author who conducted the literature search completed the title and abstract screening of the yield, along with the removal of duplicates. The review of full text articles was performed by four authors.

## 3. Results

From the initial literature search, a total of 2748 articles were identified across 7 databases (Figure A1). Following the removal of duplicate articles, titles and abstracts were screened for our key concepts, resulting in 21 articles being selected for full text assessment and a final 7 selected for inclusion. No studies were identified in our search when (1) and (5), and (4) and (5) were combined (Figure A2). There were no CMs’ scheduling studies identified in the search conducted. All studies focused on the impact of scheduling codeine products, and this was judged as relevant to understanding more broadly what needs to be considered in the scheduling of any medicines, including CMs.

Four of the studies examined the perceptions and perspectives of pharmacists in relation to the up-scheduling of codeine. Two quantitative studies examined retrospective data from the Poison Information Centres, in Australia and Ireland. One study explored the characterization of Schedule 2, Schedule 3, and unscheduled medicines. As presented in Table 1, five themes were identified related to the purpose, attitudes, and implications.

## 4. Discussion

To our knowledge, this is the first study to explore the effects of regulatory scheduling of medicines and its impact on pharmacists’ in terms of workload or professional behaviors. This review has illustrated the impact of up-scheduling medicines, using codeine as an example, and it was conducted with a view for theoretically extrapolating to other medicines, specifically to CMs. The findings of this study suggested there was a lack of quantitative data to provide specific outcomes relating to the schedule change. The majority of the literature was comprised of exploratory qualitative studies on the perceptions and opinions of pharmacists in relation to codeine.

The intent of up-scheduling codeine was to minimize codeine misuse. When codeine was up-scheduled from Schedule 2 to Schedule 3, a pharmacist’s involvement in the supply of codeine-containing medicines became a legal requirement. Pharmacists had a legal obligation and professional responsibility to ensure that codeine use was therapeutically appropriate for a patient before permitting sale. In such circumstances, pharmacists uphold an important role as a “gatekeeper” by counseling and, if required, intervening in OTC CCP misuse. In the qualitative study by Hamer et al., it was reported that there was an increase in the workload on pharmacists due to these new legal obligations [25]. Pharmacists also noted that they became more aware about misuse as a result of the increase in patients requesting codeine [25]. Some pharmacists felt that it gave them confidence to discuss with patients any codeine-related problems [25]. Some pharmacists would attempt to address a patient’s concerns and recommend alternatives to minimize codeine use [13,23]. Other pharmacists felt that rescheduling to Schedule 3 prompted them to upskill in the area of pain management [23]. Following the second up-scheduling to Schedule 4 for any product containing codeine, it was reported that the up-scheduling actually created an opportunity for patients to openly discuss their pain issues with the pharmacist and seek non-codeine strategies to manage their pain [23,24].

Whilst CMs and codeine (which has potential for misuse) are very different classes of medicines, we propose that the fundamental principles that prompted changes to the up-scheduling of codeine are the same, i.e., pharmacists’ responsibility to engage in patient care that ensures the appropriate and safe use of medicines. Currently, CMs are categorized as unscheduled medicines, and, as such, there is no legal implications for pharmacists if they do not directly engage in the supply of CMs. However, there is growing evidence of adverse reactions between certain medicines and some CMs, which potentially have serious effects on patient safety [27,28]. Therefore, there is a gap between the regulation of supply of CM products and pharmacy personnel’s, including pharmacists’, professional responsibilities. To date, up-scheduling of CMs has not been discussed in the literature.

Shifting CMs to Schedule 2 could theoretically increase pharmacy personnel’s involvement in supply. When codeine was up-scheduled to “Pharmacist-only medicine”, Hamer et al. reported that pharmacists’ awareness about patients’ request for codeine increased, which prompted for identification of codeine misuse and changes in pharmacists’ practices with regard to supplying of CCPs [25]. Again, while CMs are overall a lower risk than CCPs, they are not without risk.

Therefore, we hypothesize that, by having an established legal framework, subjecting CMs to up-scheduling to at least Schedule 2, and educating pharmacy assistants, technicians, and pharmacists in the handling of CMs will enable closer monitoring [11] of CM use and identification of drug–herb interactions. With heightened regulation, pharmacy personnel would be required by law to interact and engage with the patients during the sale of CMs. This will create an opportunity for patients to make more informed decisions about the CMs of interest and encourage appropriate and safe use. At the same time, having CMs available exclusively through the pharmacy could be a strong message to consumers to be aware of and comply with advice offered with respect to the safety of CMs, as they would have with other medicines or prescription medicines. Consumers’ perceived need to consult with pharmacy personnel when deciding on the use of CMs would, therefore, also be prompted. As indicated by a consumer advocacy group representative, “*it would be ideal to have pharmacists to deal with consumers’ use of CMs*”. Initiating the regulatory scheduling of CMs might be the remedial policy needed to correct consumers’ belief systems about the safety of CMs.

Furthermore, in a study conducted by Emmerton in 2003, it was noted that Schedule 3 medicines have higher rates of in-store interventions and pharmacist engagement, as compared to unscheduled medicines [8]. A probable conclusion drawn from this study could be that medicines assigned with higher risk value would be prioritized by pharmacy staff [13]. While CMs are commonly perceived by the public to be safe medicines [5], adverse reactions, drug-herb interactions and serious toxicities, if used inappropriately have been reported [27,28]. Increased risk of drug and CMs interaction is present in physiologically-compromised individuals and the elderly that are on polypharmacy [29,30]. Up-scheduling of CMs can encourage more patient-pharmacist/pharmacist technician interactions on the use of CMs. This may also potentially increase the reporting of drug-herb interactions and raise awareness among CM users and pharmacy personnel about interactions and, even more importantly, add to building the evidence-base about CM safety.

Another significant finding of this study was that one of the major challenges reported by pharmacists with the scheduling of codeine to Schedule 3 was having to establish a “therapeutic need” as required by law. Due to time constraints in real-time pharmacy practice, pharmacists felt that they were unable to perform a detailed consultation session with patients to determine a therapeutic need [13]. Based on the codeine example, we might be able to extrapolate that pharmacists might see it as a challenge to conduct a comprehensive consultation with their patients on their conventional medications and CMs, with time being the limiting factor. This provides an opportunity for pharmacy assistants/technicians to extend their role and be trained in CMs counseling. Doucette reported that pharmacy technicians were willing to perform new tasks that are needed to support emerging patient care services within community pharmacies [31]. The need for pharmacist assistants’ or technicians’ roles to evolve through appropriate education and collaborative working relations with pharmacists has been suggested [32]. Such an evolution would be suited to filling the current gap in the provision of professional care related to CMs use.

Despite legal obligations for the supply of Schedule 3 codeine, there was a reported lack of conformity between Australian pharmacies [13]. Inconsistencies in pharmacists’ practices were found in different pharmacies, which can also become a potential barrier between patient and pharmacist. Some patients reported that pharmacists were not able to provide the information about CMs’ they needed and therefore decided not to ask the pharmacist’s assistance in making a decisions about CMs again [11]. Other patients may become confused and interpret the behavior of pharmacists exercising a duty of care as being unnecessarily prohibitive in comparison to other pharmacists who do not exercise professional behaviors related the provision of CM products [13]. With that in view, if CMs were to be up-scheduled, we might also be faced with varying pharmacy practices in the supply of CMs. Perhaps some of this inconsistency would be reduced if Schedule 2 was applied to CMs and well-trained pharmacy assistants and technicians could ‘absorb’ the professional responsibility of the counseling session and refer to the pharmacist only when necessary.

In addition, it was reported that when codeine was regulated to Schedule 3, some pharmacists were concerned about their clinical competence not being sufficient to help patients with pain management without the aid of over-the-counter CCPs [24]. It was consistent in most of the studies that the up-scheduling of codeine required additional educational training for pharmacists in the area of pain management, counseling skills, and the handling of drug misuse [13,23,24]. This could imply that, if a change in CMs scheduling were to eventuate, pharmacists, pharmacy assistants, and pharmacy technicians would need to undertake professional training in the area of CMs to build on their evidence-based CMs knowledge. In the study by Ung et al., education is identified as one of the solutions to facilitate the integration of CMs into pharmacy practice [33].

We suggest that up-scheduling of CMs should be based on an evidence-based model and conducted in a step-up approach. Suggestions include up-scheduling CMs that are associated with high interactions-risk and used as a therapeutic agent. Some Australian CMs companies have attempted to promote practitioner-only CMs. Yet little is known about the impact on improving pharmacy personnel engagement with consumers who are buying these products. With that in mind, scheduling of medicines should be explored in depth in relation to the possible implications for other healthcare professionals, retail outlets owners, and, more specifically, CMs practitioners. The aim of such research would be to promote a coordinated approach to supporting the appropriate and safe use of CMs across different sectors.

This study does have some limitations. As this is a newly emerging topic, there is limited research to date and lack of sufficient opportunity to compare with previous studies. Hence, it may not be an adequately comprehensive study to fully inform readers and policy makers on the impact of rescheduling. However, the findings of our study have clearly flagged several suggested initiatives that are worthy of consideration.

As codeine is a drug of addiction, we also need to be mindful that the purpose and effects of up-scheduling could be different to the case of CMs; although patient safety is of the essence of the approach taken and clearly has become applicable to CMs. Further research to inform enhancement of the role of pharmacy personnel engagement in the sale of CMs in view of the position statement from PSA may be required.

## 5. Conclusions

The case of codeine illustrates the possible impact of regulatory up-scheduling of medicines in community pharmacy. If CMs were to be up-scheduled to a rigorous pharmacy-specific regulatory level, such as Pharmacy-only (Schedule 2), the accessibility of CMs would be restricted to the community pharmacy setting, providing more opportunities for pharmacists and pharmacy personnel to engage with a patient’s request for CMs and counsel on CM use, thereby contributing substantially to the appropriate and safe use of CMs. An added benefit would be enhancing patient or consumer awareness of any potential risks or interactions of CMs, rather than the current common belief that CMs are ‘natural’ products with no risks associated to their intake, regardless of patient history or conditions. However, careful collaboration and consideration about how such a regulatory change would impact other key-stakeholders, including other types of retailers providing these products and CM practitioners, requires a strategic and collaborative approach.

## Figures and Tables

**Table 1 pharmacy-08-00065-t001:** Systematic review of the selected studies.

Theme		Sub-themes	Key Findings	Reference
1	Purpose		To address codeine misuse	
2	Attitudes	2.1 Positive	Pharmacists were proactive in prompting discussions with patients	[23]
			Pharmacists recommended patients with appropriate medicines management	[23]
			Less addiction and toxicity were reported due to restrictions	[24]
			General Practitioners were in support of scheduling changes	[24]
			Up scheduling positively impacted the practice of community pharmacists in Australia	[13]
			Improvements to practice behaviors	[13]
		2.2 Negative	Did not solve misuse as patient shifted from “pharmacist-shopping” to “doctor-shopping”	[23]
			Some pharmacists felt that it might have possibly lead to escalation of stronger medications	[23]
			Limited pharmacists’ capacity in offering pain management	[23]
			Some pharmacists viewed up scheduling of codeine as increasing GP’s burden	[23]
			Opposition to the scheduling by pharmacists and users	
			Had a negative impacts on consumers’ health, finances and pain management	
3	Potential impact on practice	3.1 Treatment options	Concerns raised around treatment options and support for pain management after the restriction	[23]
			Establishing therapeutic needs, inconsistent supplying issue between pharmacies and intervening with codeine-dependent individuals	[13]
		3.2 Challenges	Impact on business and environmental factors	[23]
		3.3 Funding models of payment	Implications to pharmacies income	[23]
4	Experiences of impact	4.1 Positive	Resulted in a decrease in the reported poison cases involving non-prescription codeine products in 2011	[24]
			Pharmacists required to monitor supply and identify more cases of misuse	[25]
		4.2 Insignificant	Rate of codeine poisoning remained stable and at a lower level	[26]
		4.3 Negative	Had no impact on misuse; Possible reason to why Schedule 3 failed to make an impact: People misusing codeine did not necessarily fit the ‘addict’ stereotype	[14]
			Pharmacists were not confident discussing possible codeine dependence with patients	[26]
5	Related issues	5.1 Marketing and advertising	Misleading patients to think that codeine is an effective treatment for pain	[12]
		5.2 Compliance with legislation and professional guidelines	Greater staff involvement for scheduled medicines	[12]

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
