# Peer review of "Impact of Up-Scheduling Medicines on Pharmacy Personnel, Using Codeine as an Example, with Possible Adaption to Complementary Medicines: A Scoping Review"

_pharmacy, 2020, doi:10.3390/pharmacy8020065_

Round 1

Reviewer 1 Report

I am so glad to see an article about this; even in the united states, many of the CM products are available for patients to buy and many patients can buy products without any consultation from the pharmacist, which could lead to significant drug interactions, so I think that this is a concept that needs to be had worldwide. 

While your conclusion makes sense, it is a bit of a stretch, given that you are basing it on codeine. Personally, I agree with the recommendation, just speaking from an evidence-based standpoint. The roadblocks regarding the implementation of up-scheduling probably would be the same. 

Were there any studies that looked at general attitudes of pharmacists regarding CM?  Perhaps this could provide some insight into what pharmacists think about these items.

Author Response

Reviewers comments

I am so glad to see an article about this; even in the united states, many of the CM products are available for patients to buy and many patients can buy products without any consultation from the pharmacist, which could lead to significant drug interactions, so I think that this is a concept that needs to be had worldwide. 

Authors response

Thank you for your encouraging comment. We agree that this is conversation that needs to be had.

Reviewers comment

While your conclusion makes sense, it is a bit of a stretch, given that you are basing it on codeine. Personally, I agree with the recommendation, just speaking from an evidence-based standpoint. The roadblocks regarding the implementation of up-scheduling probably would be the same. 

Authors response

Thank you for your comment. We have tempered our conclusion to read:

‘The case of codeine illustrates the possible impact of regulatory up-scheduling of medicines in community pharmacy. If CMs were to be up-scheduled to a rigorous pharmacy-specific regulatory level, such as Pharmacy-only [Schedule 2], the accessibility of CMs would be restricted to the community pharmacy setting, providing more opportunities for pharmacists and pharmacy personnel to engage with a patient’s request for CMs and counsel on CM use, thereby contributing substantially to the appropriate and safe use of CMs. An added benefit would be enhancing patient or consumer awareness of any potential risks or interactions of CMs, rather than the current common belief that CMs are ‘natural’ products with no risks associated to their intake, regardless of patient history or conditions. However, careful collaboration and consideration about how such a regulatory change would impact other key-stakeholders, including other types of retailers providing these products and CM practitioners, requires a strategic and collaborative approach.’

Reviewers comment

Were there any studies that looked at general attitudes of pharmacists regarding CM?  Perhaps this could provide some insight into what pharmacists think about these items.

Authors response

Thank you for your important comment. Indeed there are a number of studies our group have published related to pharmacist’s perceptions about complementary medicines. We have summarised some of these findings and added the following text to the manuscript on page 3:

‘Despite this, there are reports that barriers exist in the pharmacy environment, that prevent pharmacists and pharmacy personnel from adopting professional standards regarding CMs. These barriers appear to be: time constraints, limited resources within the pharmacy, and a lack of knowledge about CMs [11]. Some pharmacists report that CMs were secondary to their primary concerns in patient care. They considered these products as being mostly “retail” products, available through a range of retail outlets in addition to pharmacies, indicating the uncertainty about whether they should assume professional responsibility for ensuring the appropriate and safe use of  widely available CMs [12].’

Reviewer 2 Report

The Authors point out that in Australia there is no suitable regulation for alternative medicines

The analysis of the study is correct and accurate.

why did the authors only use the codeine molecule as an example? there are no other studies on other molecules

Author Response

Dear Reviewer number 2,

Thank you for your comment. Indeed we agree the choice of codeine as an example may seem curious. We do allude to this codeine being ‘incomparable’ in many ways. However, we were unable to identify any other examples of up-scheduling in Australia. 

Thank you,

The authorship team

Reviewer 3 Report

This manuscript is a review article for up-scheduling medicine on pharmacy workforces.
The contents fits the journal's scope.

The manuscript is well organized and well-written.

The results obtained by literature review are quite intersting.

But I recommend to add some comments for following points.

Only things what I can suiggest:
1. Authors defined the CMs as complementary medicine, natural medicine . dietary supplemets etc.
Some would be used for therapeutic to diseases, others are used to maitain their helth.
It would be preferable to discuss separately even if it may not change in contribution of pharamcists and pharmacy technicians.
2. Regultaion of drugs in each countries are quite difference. I understand tt is because authors discussed mainly to use Codein preparations. Scheduing of medicines is quite depending on country by country.
Discussion based on regulation in some countries would be useful to understand international readers.

Author Response

Reviewer 3

This manuscript is a review article for up-scheduling medicine on pharmacy workforces.
The contents fits the journal's scope. The manuscript is well organized and well-written. The results obtained by literature review are quite interesting. But I recommend to add some comments for following points.
Only things what I can suggest:

Reviewer Comment
1. Authors defined the CMs as complementary medicine, natural medicine, dietary supplements etc.
Some would be used for therapeutic to diseases, others are used to maintain their health.
It would be preferable to discuss separately even if it may not change in contribution of pharmacists and pharmacy technicians.

Author’s response

Thank you for your comment. We agree there are some complementary medicines that have evidence for efficacy for the treatment of certain conditions. Therefore, we have added some text highlighting the example of St John’s Wort and the treatment of mild to moderate depression. We have also used this example to highlight the range of interactions associated with the use of this herb. The following text has been added on page 3 of the manuscript:

‘However, as they are provided from within pharmacy premises, and according to principles of professional ethics clearly articulated in the Code of Ethics for Pharmacist [6], pharmacists and pharmacy personnel are expected to counsel patients, providing sufficient information to ensure safe and appropriate use and that pharmacists adopt an engaged professional role in relation to CMs provision [4, 5]. This is particularly important in light of the growing body of evidence reporting potential side effects and drug-CM interactions that can impact on patient safety and clinical outcomes of therapy. An well-established drug-herb interaction is Hypericum perforatum (St John’s Wort)[7] which is used in the management of mild to moderate depression [25]. In Australia, St John’s Wort is easily accessed over-the-counter (OTC). Hyperforin, an active constituent of Johns Wort, is known to induce cytochrome P-3A4 (CYP3A4) enzyme and the drug transporter p-glycoprotein which are involved in the metabolism of many medications [8, 9]. Concurrent use of SJW with these CYP3A4-metabolised medicines is well documented as resulting in significant interactions associated with patient harm [10].’

Reviewer Comment
2. Regulation of drugs in each countries are quite difference. I understand it is because authors discussed mainly to use Codeine preparations. Scheduling of medicines is quite depending on country by country. Discussion based on regulation in some countries would be useful to understand international readers.

Authors response

Thank you for your comment. We appreciate that regulations for both pharmaceutical medicines varies across regions. The following text has been added to the manuscript to provide some context for the reader on page 4

‘….. is legally required in the case of ‘Pharmacist-Only’ (Schedule 3 in Australia) and ‘Prescription-Only’ (Schedule 4 in Australia) medicines [11], as well as clearly mentioned in the Code of Ethics for Pharmacists, it is unclear whether such professional responsibility could be extended to CMs in their current status or should up-scheduling be considered?’

……‘One example of the complexities and impact of rescheduling was the rescheduling in Australia of codeine-containing products (CCPs) in 2010, which was a very controversial topic at the time. Some low dose CCPs were previously scheduled at Schedule 2 or over-the-counter (OTC). However, with intense scrutiny of the misuse of opioid-containing medicines, and the rising incidence of addiction on these medicines in Australia, on the 1st of May 2010, regulatory changes were made to minimise access and misuse, and all low dose CCPs were up-scheduled from Schedule 2 to Schedule 3 [14]. In Australia, patients could therefore no longer self-select CCPs, and the pharmacist was required to have some level of consultation with the patient before supply of the product. Being a Pharmacist-Only Medicine implied that the pharmacist had all professional and legal responsibilities for the supply of the Schedule 3 medicine, and as such, up-scheduling impacted on the practice of community pharmacists all around Australia [14].